# Assessing the Use of Hospital Information Systems (HIS) to Support Decision-Making: A Cross-Sectional Study in Public Hospitals in the Huíla Health Region of Southern Angola

**DOI:** 10.3390/healthcare10071267

**Published:** 2022-07-07

**Authors:** Tomas Hambili Paulo Sanjuluca, Anabela Antunes de Almeida, Ricardo Cruz-Correia

**Affiliations:** 1Faculty of Health Sciences, University of Beira Interior, 6200-386 Covilhã, Portugal; 2Faculty of Medicine, Mandume Ya Ndemufayo University (FMUMN)—Huila-Angola, Av. Hoji ya Henda 30, Lubango P.O. Box 201, Angola; 3Department of Management and Economics, Faculty of Social and Human Sciences, University of Beira Interior, 6200-209 Covilhã, Portugal; aalmeida@ubi.pt; 4NECE—Research Unit in Business Sciences, Estr. do Sineiro 56, 6200-209 Covilhã, Portugal; 5Department of Community Medicine, Information and Decision in Health—MEDCIDS, Faculty of Medicine, University of Porto, 4200-450 Porto, Portugal; rcorreia@med.up.pt

**Keywords:** Angola, hospital management information systems, hospital information system, support decision-making, data quality

## Abstract

Hospital information systems could be relevant tools to inform hospital managers, support better management decisions in healthcare, and increase efficiency. Nonetheless, hospital managers’ effective use of these systems to support decision-making in Angola is unknown. Our study aimed to analyse the use of hospital information systems as a tool to support decision-making by hospital managers in Huíla, Angola. It was a descriptive, cross-sectional study inducted between July and September 2017 in seven hospitals in Huíla Province, Angola, specifically in the cities of Lubango and Matala. Thirty-six members of the hospital boards filled out a self-questionnaire that consisted of twenty questions based on the following issues: Characterisation of the interviewee’s profile; availability of information in the institution; and quality and usefulness of the available operational information. At least two thirds of the participants reported being unsatisfied or relatively satisfied with each assessed hospital information systems-specific feature. More than 50% have rarely or never used the health information system to support decision-making. Most managers do not use hospital information systems to support management-related decision-making in Angola. Improving the ability of hospital information systems to compute adequate indicators and training for hospital managers could be targets for future interventions to support better management-related decision-making in Angolan healthcare.

## 1. Introduction

Hospital information systems (HIS) can support the improvement of healthcare delivery, patient safety and satisfaction, and clinical practice. HIS could be a relevant tool to inform hospital managers, support better management decisions in healthcare, and, consequently, increase efficiency.

A HIS can be defined as a computerised or manual system (on paper) that is designed to meet all the information needs within a hospital. This includes different types of data (heterogeneous information), such as patient information, billing, finance and accounting, staffing, scheduling, pharmacy ordering, prescription handling, supplies, inventory, maintenance, order management, diagnostic reports related to laboratory, and patient monitoring, as well as providing decision support [1].

A health information system (HIS) is a set of components (technical, organizational, behavioural) and procedures “organized to generate information to improve health management decisions at all levels of the health system”. For example, healthcare providers collect data on health services, states, and resources. When a HIS produces high-quality, timely, and reliable data, it enables health program managers to monitor, evaluate, and improve health system performance, and to make evidence-based decisions [2,3].

Health data provide a picture of health status, health services, and resources. The sources of these data are usually records of services provided, individual medical records, and health resource records, which provide information about the health of patients and the type of treatments and tests they receive. Managers may collect other information on human resources, finances, medicines, and supply systems [4,5]. However, we can also turn to public health advisors, hospital and healthcare managers, and ongoing surveys of healthcare facilities, which also provide information. However, studies show that there is a need for governments and health decision-makers to create strategies to develop tools at the hospital, local district, and national levels to use better routine health information systems (RHIS) [6,7,8].

Various hospital information systems are used by hospitals today, which has made decision-making easier, be it the diagnosis and treatment of patients or regular hospital management functions like admission, laboratory, billing, inventory, pharmacy, finance and accounting, outpatient management, etc. However, weak HIS is a critical challenge to reaching the health-related Millennium Development Goals, because health systems performance cannot be adequately assessed or monitored where HIS data are incomplete, inaccurate, or untimely [9,10].

A health systems’ goal is to optimise the health of the treated individuals and populations, and the “gold standard” for quality measurement will thus always be health outcome measures. HIS can be defined as the sociotechnical subsystems of a hospital, comprising all information-processing systems and the associated human or technical actors in their information-processing roles [7,11,12].

Better use of information requires better quality data and products, which, in turn, require better health information systems (HIS). Health system performance depends on producing and using quality health data and information. Routine health information systems (RHIS) are defined as systems that provide information at regular intervals of one year or less to meet predictable information needs. These are defined as routine information systems, including paper or electronic health records and facility and district management information systems. RHIS is receiving increasing attention as an essential component of efficient, country-owned, and integrated national systems [6,7,13].

Studies that have analysed health information systems show that socioeconomic factors affect safety, including for people, technology, tasks, organization, and the environment. For example, there is evidence that HIS usability problems, such as patient identification, have caused user errors that led to patient safety incidents [14]. In addition, healthcare professionals’ behaviours can result in unintended consequences for HIS. Knowledge about these contextual issues can help healthcare organizations, healthcare professionals, and HIS implementers understand the in situ functioning of HIS in its use context and help them design strategies to decrease the number of unsafe HIS adoptions and their possible negative consequences [15,16].

Sound hospital information systems, computerised or not, should present timely, correct, and appropriate information to the right persons. However, there are plenty of data sources in a hospital, with many different functions with mutually complex relations, and many people are dependent on this information [9,10]. 

Given the enormous volume of data generated in hospitals, to efficiently manage them, using HIS is critical. User participation is one of the significant factors in the success of HIS, which, in turn, leads to information needs and processes being correctly predicted and their commitment to the development of HIS being augmented [17,18].

Appropriate decisions taken by hospital managers are fundamental to ensure compliance with high-quality standards of patient care. In addition, those decisions benefit from systematic information-processing that contributes to achieving the hospital’s strategic goals [19,20].

HIS is one part of a health system’s six essential and interrelated components. A well-functioning health information system should produce reliable and timely information on health determinants, health status, and health system performance and be able to analyse this information to guide activities in all the other building blocks of the health system [21,22].

Many studies assess if and how healthcare managers apply HIS in their decision processes. For example, hospital managers in Portugal considered HIS to be a relevant tool in the institutional organisation used in the decision process but only incidentally and not systematically [23,24]. Some difficulties raised by the professionals were the technical difficulty, the nonspecific system, lack of training, and the difficulty in producing effective processes to the demands of users [25,26]. These results agree with a study carried out in Brazil, the findings of which showed that ample amounts of information and data are available in public hospitals. However, it presents many gaps, as hospital managers do not know about the existing data and do not use it to guide hospital management [20]. 

Evidence shows that the work and time spent by hospital managers can influence quality and safety clinical outcomes, processes, and performance. Many issues affect quality performance, such as establishing goals and strategies to improve care, setting the quality plan, engaging in quality, and so on [27,28,29]. 

It is the responsibility of the Angolan state to promote and guarantee the access of all citizens to healthcare within the limits of available human and financial resources. The organisation of the healthcare delivery system in Angola subdivides into three hierarchical levels [30]:The primary level is represented by municipal health institutions (health centres and hospitals).The secondary or intermediate level is the reference level for first-level units, represented by provincial health institutions (general hospitals).The tertiary level is the reference for secondary level units, represented by national health institutions, such as reference hospitals, differentiated, specialised, or multipurpose hospitals.

Among other weaknesses present in the Angolan national health system, studies have identified the following as those that assume a particular relevance: (i) the difficulty of strategic articulation and coordination of health interventions and the determinants of health; (ii) weak leadership in the health sector; (iii) poor planning capacity at all levels; (iv) decentralisation without financial autonomy for local health structures; (v) poor management of resources made available at all levels; (vi) investments inconsistent with health needs and priorities; (vii) low transparency in management acts; and (viii) an incipient information, communication, supervision, and evaluation system.

The fundamental challenges to be faced when aiming to achieve a more efficient level of healthcare can be summarised in four central aspects: financing, management, access, and the qualification of professionals. To improve these points, consistent and continuous policies are needed [31]. In Angola, the direction of the hospital is performed by a board of directors composed of six directors. This board is responsible for performing all tasks related to managing hospital services, from clinical care to administration.

All information systems (IS) are implemented and maintained by the hospitals without using national information systems made available by the health ministry. Nevertheless, the Health Ministry defines a set of indicators that hospitals need to calculate and send. However, these processes are still performed using paper to both collect data and send it to the regional government and the national Health Ministry [30].

In low- and middle-income countries, health information systems have received unprecedented strengthening and attention in recent years, as evidenced by the formation of the Health Metrics Network, the convening of the Global Health Information Forum in 2010 in Bangkok, and the unveiling of President Obama’s Global Health Initiative, which calls for “strengthening existing public health surveillance and other data collection systems to monitor diseases, conditions, health service delivery, and health outcomes” as part of an integrated approach to strengthening health [4,7].

There are still significant challenges associated with the use of routine health information system data in low- and middle-income countries, but there is evidence of initiatives in some African countries that could improve data systems and the use of HIS results as a driver for health system decision-making and performance improvements [4,31,32]. 

As shown during the COVID-19 pandemic, the way healthcare is managed in one place (e.g., new variants appearing in many different countries) in the world can have a huge impact elsewhere. Therefore, a call for better healthcare management and accurate data across the globe is needed [33].

The implementation of HIS systems has increased globally over the past five years, and higher-income countries are further adapting and utilising HIS compared to lower-income countries [34].

It is essential to better evaluate the level of use of HIS and the corresponding barriers. Considering the relevance of HIS in the daily life of health professionals, institutions, and patients and meeting the current organisational and the general assumptions of HIS in hospitals in Angola, the questions are: Has HIS been used in the decision-making process by the managers of public hospitals in the Huíla health region? What are the main difficulties and possibilities for improving a hospital information system’s decision-making process?

Based on this questioning, this study aimed to analyse the use of the hospital information systems as a tool to support the decision-making process by hospital managers in Huíla Province, Angola.

## 2. Materials and Methods

In this section, we have tried to provide sufficient detail to allow a better understanding of the materials and methods used in this study so that the study can be replicated.

### 2.1. The Study Design

The study was a cross-sectional, questionnaire-based, observational study with a quantitative and qualitative approach. Data were collected from July to December 2017 in seven hospitals in Huíla, Angola. The study was authorised by the provincial Health Director of Huíla. 

### 2.2. Study Setting

Angola is organised into 18 provinces and, in the last census (held in 2014), had almost 26 million inhabitants. Luanda is the biggest province, followed by Huíla, which hosts 2.5 million inhabitants. Huíla Province includes 14 cities with a population ranging between 64 and 776 thousand citizens [35]. 

Data from 2006–2010 estimated that Angola had 2356 public healthcare institutions, including, among others, 165 municipal hospitals, 25 province hospitals, and 20 central hospitals. Huíla had the second-highest healthcare institutions (*n* = 232), just after South Kwanza (*n* = 244).

### 2.3. Hospital Selection Process

We used a two-step selection process. First, we identified the Huíla cities with more than 200,000 inhabitants. Lubango (the capital) and Matala are the cities with the most significant number of inhabitants and the only ones that meet these criteria [35].

Secondly, within those cities, we selected hospitals with at least 50 beds for inpatient hospitalisation. Lubango has five provincial hospitals and four municipal hospitals, while Matala has only one municipal Hospital. Overall, seven hospitals (78%) met the inclusion criteria (six hospitals in Lubango and one in Matala). 

### 2.4. Questionnaire

We used a self-questionnaire adapted from those described by Guimarães et al. and Cavalcante et al. [26,36]. The questionnaire had 20 questions with four main focuses: characterisation of the interviewee’s profile; availability of information in the institution; quality and usefulness of the available operational information; satisfaction with the existing HIS.

Both closed and open questions were included, depending on the target information (e.g., available questions to obtain unbiased information related to HIS difficulties and enhancement suggestions).

A pilot study was conducted with 15 Angolan students from the specialisation course of the health management program at the Professional Health School of Huíla to assess the comprehensibility and ease of filling the study’s self-questionnaire. They suggested some changes to the questions, considering the country context, which was viewed in the final questionnaire. 

### 2.5. Participants and Data Collection

After being tested and approved, the questionnaire was applied in public sector health institutions with an authorisation order for the study to be carried out signed by the Director of the Health Region of Huíla. All members of the institutional board of the selected hospitals were invited to participate in this study. Usually, the committee included four to six managers with different tasks: general, clinical, administrative, nursing, diagnostic and therapeutic, and teaching and training managers (Table 1). Within each institution, the respective hospital director was responsible for distributing the questionnaire to all managers, guiding them through filling, and clarifying any doubt. Whenever needed, the responsible study team was contacted to provide further clarification. 

### 2.6. Data Analyses

The data collected for the present study were subject to descriptive statistics interpretation and analysis by using the frequencies to describe the categorical variables presented through absolute and relative frequency tables, using the Software Statistical Package for Social Sciences (SPSS) version 21.0 as a supporting tool (IBM, Armonk, NY, USA).

### 2.7. Ethical Considerations

The study was conducted with the permission of the Director of the Health Region of Huíla. Our research study request was evaluated and approved, without number, and it signed on 14.09.2017, in Lubango by the Director. This was not direct research on human subjects. The documents examined contain private and confidential information, so we endeavoured to comply with the standards and administrative procedures established by the general and clinical management of hospitals.

## 3. Results

Thirty-six hospital managers participated in this study (Table 1). All hospitals had general, clinical, administrative, and nursing managers; six had teaching and training managers, and only two had diagnostic and therapeutic managers. 

### 3.1. Characterisation of the Interviewee’s Profile

Around three quarters of the participants were 40 years or older. Most were nurses, of which 44 had less than five years of experience in hospital management (83%). In total, 72% of the participants had attended basic computer training, 61% had training in health statistics, and 58% participated in a training program in health management. Additional participant characteristics are described in Table 2.

### 3.2. Availability of Information within the Selected Institutions

In total, 64% of the participants reported that data availability was reasonably acceptable, and 11% considered it unacceptable. Forty-four percent of the managers reported never using the available information in decision-making regarding patient care. Regarding the support for clinical or administrative decision-making, nearly 39% of the participants reported never using this support (Table 3). 

### 3.3. Quality and Usefulness of the Available Operational Information

Only 25% of the participants felt that the HIS-stored information was satisfactory in calculating the rate of service indicators, and 33% believed that the stored information did not compute indicators. 

According to 47% of the participants, there was an absence of audit sessions to assess the quality of information described in the clinical records sent to admission, archive, and health statistics services. 

These study participants reported that the most relevant limitations regarding the information access process were the lack of important/required information in the available dashboards (41%). The priority information requirement was computerising all information on hospital activities (41%) (Table 4).

### 3.4. Satisfaction with the Existing HIS

At least two thirds of the participants reported being unsatisfied or relatively satisfied with each of the HIS-specific features that were assessed (security and archive mechanisms used for clinical files, access to relevant information, sharing of information among hospital services, clinical file information content, and comprehensibility) (Table 5).

No more than 9% were delighted with any of the assessed features. Furthermore, three quarters of the participants reported that their institutions had no software project or investment policy regarding electronic health records. Nevertheless, they were unanimous in considering that these could improve hospital performance using information technology. 

### 3.5. Other Questions Related to the Study

In this section, we presented both the closed and open questions we included, which depended on the target information (e.g., available questions to obtain unbiased information related to HIS difficulties and enhancement suggestions). Finally, we presented the answers to the open questions in a summarised form. We note that the study participants expressed their concerns regarding the hospital information system in force in each of the institutions.

## 4. Discussion

This section discusses the significance of the most relevant findings of the study.

### 4.1. Statement of Principal Findings

This study analysed the reality of hospital managers in Angola’s Huíla health region regarding the use of HIS as a tool to support the decision-making process. An analysis of the questionnaires filled out by thirty-six managers from the seven hospitals that participated in the study revealed that hospital managers do not use HIS to support decision-making for clinical and administrative aspects. In addition, the participants expressed an assessment of dissatisfaction with HIS. 

### 4.2. Strengths and Limitations

The main strengths of this study are that there are very few studies on health informatics in the large region of Southern Africa, a region with more than 210 million inhabitants. Furthermore, this work provides a clear picture of an Angolan province (Huíla), covering all but military hospitals. In the current pandemic, having this picture of less-studied but very populated regions is critical to support worldwide efforts.

However, this study had some limitations. First, only Huíla Province was assessed. Although the main cities of Huíla (Lubango and Matala) are among the most populated cities in Angola, we believe that it would be interesting to include a broader sample to better picture the Angolan use of HIS for decision-making in hospital management. Moreover, in the study’s questionnaire, additional questions assessing specific characteristics of HIS use at each hospital would be helpful. Finally, we assessed HIS-related perceptions, but we cannot guarantee that all hospitals used the same specific HIS (which can drive diverging perceptions, as their features might be different). 

### 4.3. Interpretation within the Context of the Wider Literature

#### 4.3.1. Characterisation of the Interviewees’ Profiles

To the best of our knowledge, this is the first study analysing the use of HIS as a tool to support decision-making by hospital managers in Angola. This study involved the main actors responsible for hospital management (e.g., clinical, nursery, administrative manager), with extensive experience within the institution and good knowledge of the available HIS. We understand that these are vital aspects when looking for a critical view of what should be changed or enhanced in HIS, thus strengthening our findings. 

Some studies have shown that a high educational level of healthcare managers can improve patient care and data quality. However, our study noted that most respondents had a minimum level of education (bachelor’s degree), making our results less generalisable to settings where most hospital managers are highly schooled. 

Previous studies have shown that having informatic courses/training could significantly affect competency levels [30]. However, in our study, despite most of the respondents having complementary informatics training, they did not have health statistics training, which would be very important to improve their data understanding and practical applications. This lack of training may be related to the absence of specific health informatics curricula in many health-related schools (e.g., medical and nursing) [37]. 

However, we feel they should actively pursue training in health management to enable hospital managers to effectively improve their decision-making by making better use of the vast amount of data produced within each healthcare institution [28,38]. 

#### 4.3.2. Availability of Information within the Selected Institutions

We found that half of the participants had never or rarely used HIS information to support decision-making regarding hospital management in Angola. However, in other countries, such as Iran and Brazil, some studies have found that hospital managers frequently use evidence-based hospital management, and they acknowledge that HIS is a relevant tool with respect to institutional organisation and should be used during the decision-making process [10,26,36,39]. Similar results were found in a study conducted in the Brong Ahafo region in Ghana on utilising the national cluster of the District Health Information System (DHIS2) for health service decision-making at the district, subdistrict, and community levels. It was found that despite 93% of the health facilities studied submitted data to the DHIS2 platform, the evidence still suggested the low use of these data in decision-making [40,41].

Our participants reported that they did not appeal to admission, archive, and medical statistics services to examine a patient returning to an appointment after 30 days. In other words, we can say that, for patients registered for a meeting, if they return to the hospital after more than 30 days, a new admission would be performed, ignoring information collected in the previous appointment.

#### 4.3.3. Quality and Usefulness of the Available Operational Information

Otherwise, we found that, in Angola, most of our participants regarded HIS as an inappropriate tool with an excessive data volume but lacking relevant information (e.g., to compute healthcare indicators) and proper communication channels, leading to general dissatisfaction with HIS features. Although these aspects can be “true” limitations of HIS, we cannot discount that a possible lack of training in HIS might also influence hospital managers’ perceptions. Another study in Brazil found that ample information and data were available. However, managers did not know about existing data and did not use that information to guide hospital management [25].

Auditing procedures are fundamental to assess the information pathway and guarantee that recommended data collection, storage, and access procedures are fulfilled; it allows the early identification of system misuse and enables the intelligent implementation of preventive and corrective measures [42]. However, we found that almost half of the participants reported never performing information-related audit procedures in their institutions. Findings from several studies invoke the critical need for formal and informal training in health management for health managers, emphasising members of hospital boards [18,38,43]. Most hospitals in higher-income countries use comprehensive HIS, while in other parts of the world, hospital orders for medications, laboratory tests, and other services are still paper-based [34].

Another relevant aspect in the analysis of this group’s results is that 41% stated that the lack of provision of necessary information by the existing service areas at the hospital level is the main difficulty in accessing information. This confirms that the hospital information system’s weaknesses begin at the data collection stage, associated with insufficiencies in communication between the technical and managerial levels. 

Regarding the computerisation of data on hospital activities and the organisation of file systems, 69.5% of respondents considered that a priority requirement. These data express the recognition by members of the hospital directorates of the problems and weaknesses of HIS, as they unanimously pointed out which informational requirements should be prioritised for the improvement of access, treatment, and the availability of information at the hospital institution level.

There is evidence that improving the use of health information is an integral part of scaling up the provision of quality health services. Better use of information requires better quality data and products, which, in turn, requires better health information systems (HIS). Thus, a HIS enables decision-makers at all health system levels to identify progress, problems, and needs, make evidence-based decisions about health policies and programs, and optimally allocate scarce resources to achieve health improvements [21,22,44].

#### 4.3.4. Satisfaction with the Existing HIS

Moreover, hospital managers’ use of HIS seems to be influenced by factors such as age, personal motivation, work commitment, work experience, and so on [19,45]. Some of these factors might affect our findings in Angola. For example, although our study’s participants had extensive experience within their institutions, more than 80% worked in hospital management for less than five years. We did not assess personal motivation and work commitment in this study. Others studies show that longer follow-up is needed to evaluate the sustainability of programs in developing countries [11,46,47]. Other findings also support the conclusion that HIS functions in Turkish hospitals are generally not as available as quality managers would like [48].

#### 4.3.5. Other Questions Related to the Study

Regarding information technologies (IT) in hospitals, all participants in this study were unanimous in answering that the performance of hospitals could be improved with IT, which does not currently exist in their workplaces. Indeed, 75% of respondents stated that their hospitals do not have any relevant investment project or policy, such as an electronic patient records. This information reinforces the need to implement an electronic hospital information system in public health institutions. Similar results were found in a study carried out in twenty-four hospitals in the metropolitan region of São Paulo, Brazil. Only three of the interviewed directors declared that their institutions were fully computerised. In the remaining hospitals, there was a manual collection process of data [49].

Results from the systematic literature review and interviews with physicians showed that socioeconomic factors such as knowledge, system quality, information quality, service quality, training, organizational resources, teamwork, task-related stress, physical disposition, and noise influence the unsafe use of hospital information systems. Therefore, health information technologies (HIT) are recommended to reduce errors in HIS usability [50,51].

However, the weaknesses of HIS have limited managers from resorting to this essential support tool for the decision-making process. For example, the use of HIS for budget plans should be a procedure legislated by the Angolan government. Creating rules and guidelines for hospital financing based on information will create new information systems that will improve quality. This would motivate managers to strengthen and use hospital information to make clinical and administrative decisions [28,52,53]. This study stresses the need to invest in Angola’s organisation and health information technologies to gather data needed to support the decision process locally, provincially, and in the country as a whole.

Various researchers have posited that more investment is necessary for health infrastructure. HIS may contribute in different ways to quality assurance activities, such as assessing the quality of primary care, monitoring quality indicators, supporting clinical care evaluation studies, and concurrently auditing the ongoing care process using reminders or decision support techniques. However, many efforts in developing new technologies will still be necessary to meet all requests for quality assurance in real-world settings.

## 5. Conclusions

Our findings support the conclusion that most hospital managers do not use HIS as a tool to support management-related decision-making in Angola. The lack of relevant information to calculate the reported indicators is one of the significant limitations of the available information systems. Finally, further investment to improve HIS’s ability to compute adequate indicators and provide training on its usage to hospital managers across organisational levels could be targets of future interventions to support better management-related decision-making in Angolan healthcare. 

Our overview has shown that computer-based hospital systems may be a valuable tool to support a great variety of quality assurance activities in hospitals. However, we also got the impression that many efforts will still be needed to meet all the requests of persons responsible for quality assurance in different kinds of hospitals. Among others, it would be necessary to (1) improve computer-based hospital information systems to monitor quality indicators, (2) develop and use practicable outcome indicators, (3) improve methods for presenting information to health professionals, (4) develop and apply more decision-supporting techniques, (5) motivate health professionals to use hospital information systems for clinical and administrative purposes, and (6) use scientific standards to evaluate computer-assisted tool support in order to make better decisions in health organisations. 

Hospital information systems are a crucial success factor for the successful management of hospitals. The potential benefits are great, but these benefits only become accessible when the information system is used across the board.

The results of this study may lead to behavioural changes in health professionals and hospital managers due to the need to improve the quality of proper data collection and raise awareness of the importance of these data in supporting decision-making.

## Figures and Tables

**Table 1 healthcare-10-01267-t001:** Hospital managers interviewed in each organisation.

Position/ Hospital	Dr. AA Neto Central Hospital	C/da Irene Maternity Hospital	Lubango Sanatorium Hospital	Pioneiro Zeca Pediatric Hospital	Lubango Psychiatric Hospital	Matala Municipal Hospital	Lubango Municipal Hospital	Total
General manager	x	x	x	x	x	x	x	7
Clinical manager	x	x	x	x	x	x	x	7
Administrative manager	x	x	x	x	x	x	x	7
Nursery area manager	x	x	x	x	x	x	x	7
Diagnostic and therapeutic manager	x	x	-	-	-	-	-	2
Teaching and training manager	x	x	x	x	x	x	-	6
Total of managers by Hospital	6	6	5	5	5	5	4	36

(x) Answered the questionnaire; (-) did not answer the questionnaire (due to absence).

**Table 2 healthcare-10-01267-t002:** Characterisation of the participants’ profiles (*n* = 36).

Variables	*n*	%
Sex, male	18	50
Age categories, years		
≤30	3	8.3
31–40	7	19.4
41–50	19	52.8
≥50	7	19.4
Profession		
Nurse	16	44.4
Physician	15	41.7
Psychologist	3	8.3
Hospital administrative	1	2.8
Physiotherapy technician	1	2.8
Laboratory technician	1	2.8
Academic qualifications		
Bachelor’s degree	16	44.4
Specialisation	15	41.7
Master’s degree	3	8.3
Other	2	5.6
Experience in the Institution, years		
1 to 5	4	11.1
6 to 10	10	27.8
11 to 15	9	25
>15	13	36.1
Experience in hospital management (years)		
1 to 5	30	83.3
6 to 11	3	8.3
>11	3	8.3
Training courses		
Had training in informatics	26	72.2
Had health statistics training	14	38.9
Had health management program training	21	58.3

**Table 3 healthcare-10-01267-t003:** Use of information for decision-making (clinical and administrative) (*n* = 36).

*Questions*		*n*	%
How do you consider the Availability of statistical information in your Institution?	Adequate	9	25.0
	Inappropriate	*27*	*75.0*
Have you used the services of admission, archives, and medical statistics to assist a patient who returns to the appointment after more than 30 days of hospitalisation at your Institution?	Always	5	13.9
	Sometimes	12	33.3
	Rarely	3	8.3
	Never	16	44.4
Regarding the level of support for clinical or administrative decision-making, have you used the information provided by the Admissions, Archives, and Medical Statistics services?	Always	8	22.2
	Sometimes	10	27.8
	Rarely	4	11.1
	Never	14	38.9

**Table 4 healthcare-10-01267-t004:** HIS: quality and usefulness of the available operational information (*n* = 36).

*Questions*		*n*	%
In your opinion, does the information collected and routinely available in the Health Information System (HIS) allow the calculation of indicators for assessing the Quality of care provided at your Institution?	Yes, satisfied	9	25.0
	Yes, partially satisfied	15	41.7
	Not satisfied	12	33.3
There have been audit sessions on the Quality of information described in the clinical processes that go to admission, archives, and medical statistics services?	Always	5	13.9
	Sometimes	12	33.3
	Rarely	2	5.6
	Never	17	47.2
In general, what are the main difficulties in accessing information?	Excessive data volume	7	19.4
	Lack of important/required information in the available dashboards as;	15	41.7
	Shortage of IS specialists	10	27.8
	Insufficient communication channels between the technical and management levels	4	11.1
In your opinion, what informational requirements should be prioritised to improve access, treatment, and Availability of information?	The organisation of clinical file systems	11	30.6
	Computerisation of all information on hospital activities	15	41.1
	Automatic computing of indicators	10	27.8

**Table 5 healthcare-10-01267-t005:** Assessment of participants’ satisfaction with hospital information systems (*n* = 36).

*Questions*	Very Satisfied	Satisfied	Fairly Satisfied	Unsatisfied
	*n* (%)	*n* (%)	*n* (%)	*n* (%)
As a user of the Hospital Information System, how do you assess the security and archiving mechanisms of clinical processes?	1(2.8)	7(19.4)	15(41.7)	13 (36.1)
As a user of the Hospital Information System, how do you evaluate the access to information needed for your daily work?	1(2.8)	7(19.4)	11(30.6)	17(47.2)
As a user of the Hospital Information System, how do you evaluate the sharing of information in the same Institution between different services (by doctors and nurses)?	2(5.6)	7(19.4)	14(38.9)	13(36.1)
As a user of the Hospital Information System, how do you evaluate the information content of the clinical process model?	2(5.6)	9(25)	12 (33.3)	13(36.1)
As a user of the Hospital Information System, how do you evaluate the information comprehensibility of the clinical process model?	3(8.3)	9(25)	9(25)	15(41.7)

## Data Availability

Not applicable in this study. However, the study datasets used or analysed are available in the manuscript tables.

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
