# Peer review of "Assessing the Use of Hospital Information Systems (HIS) to Support Decision-Making: A Cross-Sectional Study in Public Hospitals in the Huíla Health Region of Southern Angola"

_healthcare, 2022, doi:10.3390/healthcare10071267_

Round 1

Reviewer 1 Report

I have no concern about the paper. The paper can be accepted to Healthcare Journal after proof reading and correcting grammar and punctuation errors. 

Paper title: Assessing the use of Hospital Information Systems (HIS) to support decision-making: A cross-sectional study in public hospitals in Huila, sanitary region of southern Angola.

Abstract: Hospital information systems , could be relevant tools to inform hospital managers, sup-port better management decisions in healthcare, and, consequently, increase efficiency. Nonetheless, hospital managers' effective use of these systems to support decision-making in Angola is un-known. Study aimed to analyse the use of hospital information systems, as a tool to support decision-making by hospital managers in Huila, Angola. Is a descriptive cross-sectional study inducted between July and September 2017 in seven hospitals in Huila Province, Angola, specifically in Lubango and Matala cities. Thirty-six members of the hospital boards filled out a self-questionnaire that consisted of twenty questions based on the following issues: characterization of the interviewee's profile; availability of information in the institution; quality and usefulness of the available operational information. At least â…” of the participants reported being unsatisfied or relatively satisfied with each of the assessed hospital information systems-specific features. More than 50%have rarely or never used the health information system to support decision-making. Most hospital managers don't use hospital information systems as tools to support management related decision-making in Angola. Improving the Hospital information systems, ability to compute adequate indicators, and training to hospital managers could be targets of future interventions to support better management-related decision-making in Angolan healthcare.

Author Response

We have made a linguistic revision, although we recognize that some details are still necessary.

Thank You

Reviewer 2 Report

Nice written paper, needs some minor revision in english

Author Response

We have made a linguistic revision, although we recognize that some details are still necessary.

Thank you

Reviewer 3 Report

The aim of the research is to analyze the use of hospital information systems as a tool supporting decision-making by hospital managers in southern Angola (Africa). The subject of the paper is important, but the results may not be up-to-date. The research study request was signed in 2017, and when were the surveys of hospital managers conducted? Also in 2017? If so, do the authors plan to conduct current research? This is interesting because you can observe the changes taking place in the system.

The authors conducted a survey among hospital employees, in which they collected information on how to access and use Hospital Information Systems. It can be seen that the authors put a lot of effort into this work. As a result, they identified factors that affect the quality of work and the quality of patient service in hospitals. These are important conclusions, but have they been presented to the managers of these hospitals? Has it improved anything? They should also be directed at politicians in that country.

The paper is linguistically and technically well written. The structure of the paper is correct and contains all the essential elements, but a good literature review is missing. Have other researchers tackled a similar problem? This article should provide a broader overview of other works.

The methodology is clearly explained. The conclusions from the research are correct and very important for both patients and hospital staff.

I also have a few minor comments about the look of this article:

1. In sections 2 and 4 there is no short description between points 2 and 2.1 and 4 and 4.1 as done in section 3.

2. Spaces are missing in many places, e.g. untimely[2, 3] -> untimely [2, 3] (and other references),  (DHIMS2)for -> (DHIMS2) for

3. Tables 1, 2 and 3 there are dashes, and tables 4 and 5 do not - it needs to be unified.

Author Response

Comments and Suggestions for Authors

The research aims to analyze the use of hospital information systems as a tool supporting decision-making by hospital managers in southern Angola (Africa). The paper's subject is essential, but the results may not be up-to-date. For example, the research study request was signed in 2017, and when were the surveys of hospital managers conducted? Also, in 2017? If so, do the authors plan to conduct current research? This is interesting because you can observe the changes taking place in the system.

From the preparation of the study until the authorization was signed (July to September 2017), information was collected from the date of the study authorization until April 2018. To assess whether the situation has evolved, the authors have planned to conduct further research starting in July 2022 using the same collection instrument, in the same institutions and health, and, of course, with new managers.

The authors surveyed hospital employees, collecting information on accessing and using Hospital Information Systems. It can be seen that the authors put much effort into this work. As a result, they identified factors that affect the quality of work and the quality of patient service in hospitals. These are basic conclusions, but have they been presented to the managers of these hospitals? Has it improved anything? They should also be directed at politicians in that country.

A report with the study results was sent to the managers of the institutions participating in the study and to the director of the health region of Huila. According to the mandates, there is always a change of managers. We plan to investigate in July of this year to evaluate if anything has improved in this institution. We think that the publication of this article may be an opportunity to awaken the managers of health institutions in Angola, as well as decision-makers, technicians, and politicians.

The paper is linguistically and technically well written. The structure of the paper is correct and contains all the essential elements, but a good literature review is missing. Have other researchers tackled a similar problem? This article should provide a broader overview of other works.

In Angola, we did not find any studies addressing similar problems. However, countries like Brazil and Portugal have addressed problems related to management in public hospitals. Therefore, we seek to consult, but similar research works mainly in countries with similar realities to Angola. However, we limit the number of references present in the manuscript.

The methodology is clearly explained. The conclusions from the research are correct and very important for patients and hospital staff.

I also have a few minor comments about the look of this article:

  1. First, in sections 2 and 4, there is no short description between points 2 and 2.1 and 4 and 4.1 as done in section 3.

These changes have been made and are identified in the manuscript in yellow.

  1. Spaces are missing in many places, e.g. untimely[2, 3] -> untimely [2, 3] (and other references),  (DHIMS2)for -> (DHIMS2) for

A complete manuscript revision has been made to eliminate these missing spaces.

  1. Tables 1, 2, and 3 have dashes, and tables 4 and 5 do not - it needs to be unified.

The changes have been made, and we have unified the style of the tables.

Reviewer 4 Report

The following revisions are required.

  1. In literature review, add 3 to five more relevant and latest techniques.
  2. Add Comparison table at the end of section 1 or 2 and compare with at least 10 to 15 techniques with appropriate parameters.
  3. Please make sure your paper has necessary language proof-reading.

Author Response

Comments and Suggestions for Authors: The following revisions are required.

  1. Add 3 to five more relevant and latest techniques in the literature review.

We reviewed the literature and reinforced the introduction with the necessary background of some relevant references. Below are the paragraphs with the appropriate citations and references were improved in the introduction.

A health information system (HIS) is a set of components (technical, organizational, behavioural) and procedures "organized to generate information to improve health management decisions at all levels of the health system". For example, health care providers collect data on health services, states, and resources. When a HIS produces high-quality, timely, and reliable data, it enables health program managers to monitor, evaluate, improve health system performance, and make evidence-based decisions [1,2].

Health data provide a picture of health status, health services, and resources. The sources of this data are usually records of services provided, individual medical records, and health resource records, which provide information about the health of patients and the type of treatments and tests they receive. Managers may collect other information on human resources, finances, medicines, and supply systems[3, 4]. However, we can also turn to public health advisors, hospital and healthcare managers, and ongoing surveys of healthcare facilities, which also provide information. But studies show that there is a need for governments and health decision-makers to create strategies to develop tools at the hospital, local district, and national levels to use better Routine Health Information Systems (RHIS)[5, 6]

Better use of information requires better quality data and products, which in turn requires better health information systems (HIS). Health system performance depends on producing and using quality health data and information. Routine health information systems (RHIS) are defined as systems that provide information at regular intervals of one year or less to meet predictable information needs. These are defined as routine information systems. Include paper or electronic health records and facility and district management information systems. The HRIS is receiving increasing attention as an essential component of efficient, country-owned, and integrated national systems[5-7].

Studies that have analyzed health information systems show that socio-economic factors affect safety, including people, technology, tasks, organization, and the environment. For example, there is evidence that HIS usability problems, such as patient identification, caused user errors that led to patient safety incidents [8]. In addition, healthcare professionals' behaviours can result in unintended consequences of HIS. Knowledge about these contextual issues can help healthcare organizations, healthcare professionals, and HIS implementers understand the in-situ functioning of HIS in its use context and help design strategies to decrease the number of unsafe HIS adoptions and their possible negative consequences[9, 10].

Results from the systematic literature review and interviews with physicians showed that socio-economic factors such as knowledge, system quality, information quality, service quality, training, organizational resources, teamwork, task-related stress, physical disposition, and noise influence the unsafe use of hospital information systems. Therefore, Health Information technologies (HIT) is recommended to reduce errors in HIS usability [11, 12].

Health Information Systems (HIS) are one of a health system's six essential and interrelated components. A well-functioning Health Information System should produce reliable and timely information on health determinants, health status, and health system performance and be able to analyze this information to guide activities in all the other building blocks of the health system [2, 13, 14].

There is evidence that Improving the use of health information is an integral part of scaling up the provision of quality health services. Better use of information requires better quality data and products, which in turn requires better health information systems (HIS). Thus, a HIS enables decision-makers at all health system levels to identify progress, problems, and needs, make evidence-based decisions about health policies and programs; optimally allocate scarce resources to achieve health improvements[13-15].

In low- and middle-income countries, health information systems have received unprecedented strengthening and attention in recent years, as evidenced by the formation of the Health Metrics Network, the convening of the Global Health Information Forum in 2010 in Bangkok, and the unveiling of President Obama's Global Health Initiative, which calls for "strengthening existing public health surveillance and other data collection systems to monitor diseases, conditions, health service delivery, and health outcomes" as part of an integrated approach to strengthening health[3,6].

There are still significant challenges associated with the use of routine health information systems data in low- and middle-income countries, but there is evidence of initiatives in some African countries to improve data systems and the use of HIS results as a driver for health system decision-making and performance improvements [3,4,16]

  1. Add a Comparison table at the end of section 1 or 2 and compare with at least 10 to 15 techniques with appropriate parameters.

We do not understand which techniques the editor is referring to, but we think improving the introduction may help. For example, we think it would not be appropriate to put a table in sections 1 (introduction) or 2 (material and methods), but we are open to more details to improve the manuscript.

  1. Please make sure your paper has the necessary language proofreading.

We have made a linguistic revision, although we recognize that some details are still necessary.

References

  1. Lippeveld, T., et al., Design and implementation of health information systems. 2000: World Health Organization.
  2. Wijayati, A.T. and A. Achadi. Factors Affecting the Success of Hospital Management Information System: A Systematic Review. in 6th International Conference on Public Health 2019. Sebelas Maret University.
  3. Hoxha, K., et al., Understanding the challenges associated with using data from routine health information systems in low-and middle-income countries: A systematic review. Health Information Management Journal, 2020: p. 1833358320928729.
  4. Mbondji, P.E., et al., Resources, indicators, data management, dissemination and use in health information systems in sub-Saharan Africa: results of a questionnaire-based survey. Journal of the Royal Society of Medicine, 2014. 107(1_suppl): p. 28-33.
  5. Saigí-Rubió, F., et al., Routine health information systems in the European context: A systematic review of systematic reviews. International journal of environmental research and public health, 2021. 18(9): p. 4622.
  6. Wagenaar, B.H., et al., Using routine health information systems for well-designed health evaluations in low-and middle-income countries.Health policy and planning, 2016. 31(1): p. 129-135.
  7. Baughman, L. and S. Nu. Keys to health systems integration, sustainability, and country ownership. in PEPFAR conference, Cape Town. 2011.
  8. Balaraman, P. and K. Kosalram, E-Hospital Management & Hospital Information Systems-Changing Trends. International Journal of Information Engineering & Electronic Business, 2013. 5(1).
  9. Salahuddin, L., et al., Healthcare practitioner behaviours that influence unsafe use of hospital information systems. Health informatics journal, 2020. 26(1): p. 420-434.
  10. Hautamäki, E., U.-M. Kinnunen, and S. Palojoki, Health information systems' usability-related errors in patient safety incidents. 2017.
  11. Haverinen, J., et al., How to improve communication using technology in emergency medical services? A case study from Finland. Finnish Journal of eHealth and eWelfare, 2018. 10(4): p. 339–353-339–353.
  12. Salahuddin, L., et al., Sociotechnical factors influencing unsafe use of hospital information systems: A qualitative study in Malaysian government hospitals. Health informatics journal, 2019. 25(4): p. 1358-1372.
  13. Carvalho, J.V., et al., A Maturity model for hospital information systems. Journal of Business Research, 2019. 94: p. 388-399.
  14. Lenny, P. and S. Kridanto. Analysis of user acceptance, service quality, and customer satisfaction of hospital management information system. in Journal of Physics: Conference Series. 2019. IOP Publishing.
  15. Rochmah, T.N., M.N. Fakhruzzaman, and T. Yustiawan, Hospital staff acceptance toward management information systems in Indonesia.Health Policy and Technology, 2020. 9(3): p. 268-270.
  16. Mutale, W., et al., Improving health information systems for decision making across five sub-Saharan African countries: Implementation strategies from the African Health Initiative. BMC health services research, 2013. 13(2): p. 1-12.
